# Should the Faecal Microbiota Composition Be Determined to Certify a Faecal Donor?

**DOI:** 10.3390/diagnostics14232635

**Published:** 2024-11-22

**Authors:** Celia Morales, Luna Ballestero, Patricia del Río, Raquel Barbero-Herranz, Leticia Olavarrieta, Leticia Gómez-Artíguez, Javier Galeano, José Avendaño-Ortiz, Juan Basterra, Rosa del Campo

**Affiliations:** 1Mikrobiomik Healthcare Company, 48160 Vizcaya, Spain; cmorales@mikrobiomik.net (C.M.); pdelrio@mikrobiomik.net (P.d.R.); leticiagomezartiguez@gmail.com (L.G.-A.); jbasterra@mikrobiomik.net (J.B.); 2Servicio de Microbiología, Instituto Ramón y Cajal de Investigaciones Sanitarias (IRYCIS), Hospital Universitario Ramón y Cajal, 28034 Madrid, Spain; luna.ballestero@gmail.com (L.B.); rql1898@gmail.com (R.B.-H.); joseavenort@gmail.com (J.A.-O.); 3Unidad Central de Apoyo (UCA-GT), Instituto Ramón y Cajal de Investigaciones Sanitarias (IRYCIS), Hospital Universitario Ramón y Cajal, 28034 Madrid, Spain; uca-gt@irycis.org; 4Grupo de Sistemas Complejos, Universidad Politécnica de Madrid, 28040 Madrid, Spain; galeanojav@gmail.com; 5Ciber en Enfermedades Infecciosas CIBERINFEC, Instituto de Salud Carlos III, 28029 Madrid, Spain; 6Facultad de Ciencias de la Salud, Universidad Alfonso X El Sabio, 28691 Villanueva de la Cañada, Spain

**Keywords:** faecal microbiota transplantation, human microbiome, 16S rDNA

## Abstract

Background/Objectives: Faecal microbiota transplantation (FMT) is considered a safe and effective therapy for recurrent *Clostridioides difficile* infection. It is the only current clinical indication for this technique, although numerous clinical research studies and trials propose its potential usefulness for treating other pathologies. Donor selection is a very rigorous process, based on a personal lifestyle interview and the absence of known pathogens in faeces and serum, leading to only a few volunteers finally achieving the corresponding certification. However, despite the high amount of data generated from the ongoing research studies relating microbiota and health, there is not yet a consensus defining what is a “healthy” microbiota. To date, knowledge of the composition of the microbiota is not a requirement to be a faecal donor. The aim of this work was to evaluate whether the analysis of the composition of the microbiota by massive sequencing of 16S rDNA could be useful in the selection of the faecal donors. Methods: Samples from 10 certified donors from Mikrobiomik Healthcare Company were collected and sequenced using 16S rDNA in a MiSeq (Illumina) platform. Alpha (Chao1 and Shannon indices) and beta diversity (Bray–Curtis) were performed using the bioinformatic web server Microbiome Analyst. The differences in microbial composition at the genera and phyla levels among the donors were evaluated. Results: The microbial diversity metric by alpha diversity indexes showed that most donors exhibited a similar microbial diversity and richness, whereas beta diversity by 16S rDNA sequencing revealed significant inter-donor differences, with a more stable microbial composition over time in some donors. The phyla *Bacillota* and *Bacteroidota* were predominant in all donors, while the density of other phyla, such as *Actinomycota* and *Pseudomonota*, varied among individuals. Each donor exhibited a characteristic genera distribution pattern; however, it was possible to define a microbiome core consisting of the genera *Agathobacter, Eubacterium, Bacteroides, Clostridia* UCG-014 and *Akkermansia.* Conclusions: The results suggest that donor certification does not need to rely exclusively on their microbiota composition, as it is unique to each donor. While one donor showed greater microbial diversity and richness, clear criteria for microbial normality and health have yet to be established. Therefore, donor certification should focus more on clinical and lifestyle aspects.

## 1. Introduction

The clinical use of faecal microbiota transplantation (FMT) to treat recurrent infection of *Clostridioides difficile* (rCDI) has increased exponentially worldwide since 2013, when it was included in clinical practice guidelines for the management of *Clostridioides difficile* infection of the American College of Gastroenterology (ACG) [1], the European Society of Clinical Microbiology and Infectious Diseases (ESCMID) in 2014 [2] and, finally, the Infectious Diseases Society of America in conjunction with the Society for Healthcare Epidemiology of America (IDSA/SHEA) in 2017 [3].

Despite initial high expectations, rCDI continues to be the only approved indication for FMT, mainly for equal or greater than second recurrences [4,5,6]. FMT utility has been explored in other pathologies, such as inflammatory bowel diseases [7], metabolic diseases [8], decolonisation of multidrug-resistant bacteria [9] or reduction in autism symptoms [10], without solid scientific evidence of its benefits [11]. Consequently, the therapeutic scope of FMT is still limited for clinical applications.

The consensus around FMT’s effectiveness in rCDI is strong, with reported success rates of approximately 90% [12,13]. However, despite these promising outcomes, FMT’s widespread application faces significant barriers. The most limiting factors are both donor recruitment and the methodological processing of samples [14]. Stool banks can reduce the problem of donor availability, with a guideline to homogenise donor selection criteria [15]. In addition, in many countries, FMT must be manufactured under appropriate drug legislation, but this is not easily applicable to the in-house manufacturing in hospital facilities. This complex situation has led to the creation of specific companies, such as Mikrobiomik (https://mikrobiomik.net/en/, accessed on 21 July 2023), a biopharmaceutical company created for the research, development and production of biological drugs based on the human intestinal microbiota obtained through donations by anonymous healthy volunteers.

The donor validation process is highly rigorous. It begins with a thorough interview to assess the candidate’s medical history and lifestyle habits, followed by extensive blood and stool tests to exclude transmissible diseases. The optimal age of the donors is less than 50 years with no gender preferences, considering that older age is not advisable since aging has been associated with alterations in the composition of the intestinal microbiota. In addition, candidates with a body mass index >30 are excluded by the possible alterations of the microbiota due to obesity or metabolic disorders [15,16]. Blood and faecal samples undergo screening for viruses, parasites and pathogenic bacteria. In stool samples, it is mandatory to test for antibiotic-resistant bacteria, including methicillin-resistant *Staphylococcus aureus*, vancomycin-resistant *Enterococcocus*, extended-spectrum beta-lactamase-producing *Enterobacteriaceae* and carbapenemase-producing *Enterobacteriaceae* [14]. These laboratory tests are essential for preventing undesirable adverse events, such as the two cases of bacteraemia caused by drug-resistant *Escherichia coli* transmitted through FMT in 2019 in the United States of America [17]. According to this, the selection process is certainly strict, and finally, the certification of candidates is quite difficult.

In addition to the low number of validated donors, donor adherence presents another challenge for FMT programs. In fact, donor availability is so crucial that being unable to donate several times a week could be considered an exclusion criterion. Once a donor is certified, the period for donation ranges from 3 to 4 months, and they can even be quarantined before release as an additional security measure [15].

At present, the determination of the gut microbiota composition at the genera level is not an indispensable requirement to certify faecal donors. Initially, the characterisation of the microbiota was limited by the fact that most of the bacteria comprising the microbiota are unculturable and by the labour-intensive nature of traditional techniques. However, advances in molecular technologies, such as 16S rDNA sequencing, have significantly facilitated such studies. Shotgun sequencing can detect viruses, fungi and parasites as well as bacteria, but subsequent bioinformatics analysis is not as advanced as in the case of 16s rDNA. Although 16S rDNA sequencing is considered the gold standard for identifying microbiota composition [18,19] and has become more affordable, it is not routinely used during the donor certification process. Tools such as MicrobiomeAnalyst 2.0 (https://www.microbiomeanalyst.ca/, accessed on 21 July 2023), a free bioinformatics platform, can assist in processing sequencing data, making the technology more accessible. MicrobiomeAnalyst 2.0 supports raw amplicon sequence processing, statistical analysis and functional interpretation [20]. Compared to QIIME2 [21], one of the most widely used tools for sequence analysis that requires command-line expertise, MicrobiomeAnalyst simplifies the analysis of 16S rDNA sequences.

Over the last 15 years, a large amount of knowledge has been generated about gut microbiota. However, it has not been possible to establish either the cut-off points for defining healthy microbiota or the specific associations between certain bacteria in both healthy and disease states. There is speculation that this consensus may never be fully reached due to the great variability observed between individuals, as the structure of the microbiome depends on a wide range of genetic, clinical, nutritional, environmental and biogeographical factors. This is why researchers are now focusing on the ecosystem functionality rather than the microbiological composition, but once again, no cut-off points of normality are available [22,23]. Without knowing what constitutes a healthy microbiota, the introduction of therapeutic interventions based on microbiota modulation, such as FMT, remains complex and challenging.

At present, it is impossible to select good faecal donors based exclusively on their microbiota composition and/or functionality. The term “superdonor” has been associated with significantly satisfactory results in FMT in the case of inflammatory bowel disease, especially in ulcerative colitis assays This is not the case for rCDI, where the donor profile has no impact on the clinical efficacy [24,25]. Moreover, previous studies have reported that, immediately after FMT, the recipient has a similar gut composition to that of the donor [26,27], although after a short period of time, the patient recovers their “native” microbiota with their specific composition and distribution.

Given these challenges, more research is needed to improve donor certification. Herein, we set out to analyse the microbiota composition of different sample numbers from 10 accredited donors. Our study aimed to assess whether a detailed analysis of faecal donor microbiota composition via 16S rDNA sequencing can refine the donor selection criteria.

## 2. Materials and Methods

Faecal donor’s accreditation process. Ten faecal donors (seven males and three females) aged 18–43 years with a body mass index (BMI) below 30 were recruited for this study. All donors had previously been certified as suitable by Mikrobiomik Healthcare Company. This certification involved passing an initial screening that included a detailed questionnaire about lifestyle and medical history. To increase the likelihood of finding donors, a challenging process, the inclusion criteria were set as follows: being between 18 and 50 years old, having a BMI below 30, and being able to donate 2–5 times per week. Only the donors who successfully passed the questionnaire were then subjected to two rounds of blood and faecal analyses to rule out the presence of known pathogens. The first round was conducted prior to their initial faecal donation, and the second took place 21 days after their last donation to minimise the risk of undetected pathogens, particularly those present at concentrations below the technical threshold for detectability. Each donor contributed several faecal samples, ranging from 4 to 13, collected at different time points, with a median value of 7 samples per patient. The characteristics of the donors, including their age, gender, body mass index and the number of samples, are detailed in Table 1.

16S rDNA massive sequencing and bioinformatic analysis. DNA samples were obtained from fresh faecal samples with the QiaAMP kit (Qiagen, Hilden, Germany). Massive sequencing of 16S rDNA was conducted after selective amplification of the variable regions V3 and V4 of the 16 rRNA using the Miseq platform (Illumina Inc., San Diego, CA, USA) with a read length of 2 × 300 bp, using the primers and conditions previously reported by Klindworth [28]. The sequences with low quality, quimeric or small size were filtered using the DADA2 algorithm [29] and assigning the resulting features as representative amplicon sequence variants (ASVs). Classification was performed using the SILVA 138 sequence classifier [30]. The web server MicrobiomeAnalyst was used to study the microbial composition of the donors using default parameters, specifying that we use SILVA for taxonomical classification. The alpha diversity was estimated using the Shannon and Chao1 indexes, and beta diversity was estimated by the Bray–Curtis index. Graphical representations of community profiling (alpha and beta diversity and core microbiome), phyla and genera distribution and microbial patterns (heatmap) in different donors’ samples were obtained through this bioinformatic platform.

## 3. Results

### 3.1. Microbiota Diversity

The alpha diversity indexes obtained from all samples are illustrated in Figure 1, which gives an overview of the microbial richness and diversity of the donors. Significant differences in alpha diversity values were found using the Kruskal–Wallis statistical test for both indices (*p*-value < 0.001). These differences were characterised by lower richness in donor 23 according to the Shannon index, while donors 17 and 29 had higher richness and diversity as indicated by both the Shannon and Chao 1 indexes.

Beta diversity analysis of the donors and samples was performed using the Bray–Curtis index, which allows for the comparison of microbial community composition across different samples. Significant differences in beta diversity were identified using the PERMANOVA statistical test (*p*-value < 0.001). The results of the principal coordinate analysis (PCoA) (Figure 2) indicate that, while each donor has a distinct microbial ecosystem, there are some shared features in their communities. Donors 23 and 29 were the only donors whose microbial composition differed significantly from that of the other donors.

The stability of the donors’ microbiota was evaluated by analysing the variations in the microbial composition across their different faecal samples. Some variability was observed among the samples collected from the same donor, with donor 12 being particularly notable. Among the ten donors, only donors 23 and 29 maintained a stable microbial composition with minimal variation within their own samples.

### 3.2. Microbiota Composition

Phyla distribution among the faecal samples from the donors is shown in Figure 3. The phylum *Bacillota*, formerly known as *Firmicutes*, was the most predominant, with a median relative abundance of 64.5% (range 37–91%). *Bacteroidota*, previously referred to as *Bacteroidetes*, was the second most abundant phylum, with a median relative abundance of 16.7% (range from 1% to 50%). Only three donors (D9, D18 and D23) exhibited a lower *Bacteroidota* relative abundance, recording values between 2% and 5%. Other important phyla, such as *Actynomycota* (median relative abundance 6.7%, range 0.6–34%), *Verrucomicrobiota* (median relative abundance 6.7%, range 0–20%) and *Pseudomonota* (formerly *Proteobacteria*) (median relative abundance 2.9%, range 0.06–11%), were detected in all donors. Smaller percentages of other bacterial phyla were also present. There was notable inter-donor variability in the relative proportions of these main phyla and others. Donors 17, 23 and 29 displayed stabilities in their phyla composition with differences in diversity. Donors 17 and 29 appeared to have greater phyla diversity than 23. Additionally, there was some variability in phyla abundance in samples from the same donor.

As shown in Figure 4A, the absolute and relative abundances of bacterial genera are visualised as stacked bar plots for each sample across the donors. The analysis demonstrates considerable variability in the relative abundance of each identified genus among the different donors and within their individual samples. Agathobacter (median relative abundance of 13.35%, range 0–15.89%), Eubacterium (median relative abundance of 11.06%, range 0.87–35.26%) and Bacteroides (median relative abundance of 6.71%, range 0.39–19.63%) are among the most dominant genus in many donors and samples, with Prevotella particularly predominant in donor 13. Donors 23 and 29 seem to have a similar and more stable microbiota with fewer variations between their samples. Similarly, total abundance patterns showed that certain genus, such as Agathobacter and Eubacterium, maintain high relative abundance in some donors and samples.

Despite the fluctuations in genera distribution described, we successfully defined the core microbiome based on the relative abundance of the main genus and the prevalence of this bacterial group (Figure 5). The microbiome core consists of the three predominant genera in all donors, Agathobacter, Eubacterium and Bacteroides (mean relative abundance and range shown above), followed by Clostridia UCG-014 (median relative abundance of 6.32%, range 0.76–14.92%) and Akkermansia (median relative abundance of 6.04%, range 0.6–13.74%). All these genera show high prevalence even at low levels of relative abundance, meaning they are present in most donors, although in small quantities.

To visualise the association between the microbiota composition of the ten certified donors recruited for this study, a hierarchical clustering with a heat map of bacterial genera is shown in Figure 6. This representation illustrates the relative abundances of different genera across all samples Each donor had a characteristic compositional pattern that was clearly distinguishable from the other. Only donors 23 and 29 showed compositional patterns more similar and stable across time.

## 4. Discussion

FMT is a safe and effective treatment for rCDI that is not responding to the standard of care antibiotics according to the 2021 updates of the main clinical guidelines for the management of this type of infection [5,6]. The aim of FMT is to restore gut microbiota alterations, which is in contrast to antibiotic treatment that acts as a disruptor. However, the concrete stool component that restores microbiome competence has not been identified.

Currently, in most European countries, faeces are classified and defined as a substance of human origin (SoHO) along with blood, tissues, cells and breast milk. All these substances are submitted to a specific regulation to increase their safety and quality that will be active in a few months (https://ec.europa.eu/commission/presscorner/detail/es/ip_23_6590, accessed on 21 July 2023). Despite the quality and traceability requirements of faeces to be considered as a pharmaceutical product, the validation process for faecal donors is based solely on their lifestyle and the absence of known pathogens. Nowadays, no mandatory standards for gender, human genetics or gut microbiota composition are defined. Ultimately, there are no criteria for matching donors and recipients in the FMT process.

Finding adequate faecal donors is quite difficult; in fact, different faecal banks have reported that only 2–5% of the candidates complete the selection process [31,32]. Initially, donors were chosen from among the relatives or friends of the recipient. This practice required the certification of numerous donors, one per patient, which consumed significant resources and time. In contrast, the use of a single, anonymous, validated donor can be used for numerous recipients. This not only streamlines the donor selection process but also helps to broaden the pool of available donors, thus increasing the chances of finding suitable candidates. A systematic review in 2013 [33] evaluated the efficacy of FMT in 273 patients diagnosed with rCDI across 11 different studies. The review reported a remarkable 90% success rate for FMT. Most importantly, it found no significant difference in treatment efficacy between related and anonymous donors. This finding was further supported by a study involving 1999 CDI patients and 28 FMT donors, which also reported no donor-specific effects [34]. In our case, the donors studied have been used in real rCDI treatment situations, with no differences observed between them. These results suggest that donor selection should prioritise other criteria rather than their relationship to the recipient.

The results shown above coupled with the observation that frozen and fresh faeces have the same effect and safety [35,36] gave rise to numerous stool banks and biopharmaceutical companies focused on the development of drugs based on the human microbiome. The establishment of these types of services is an optimal approach to standardise the FMT process, ensuring stool availability on demand.

The role of Mikrobiomik is focused on the investigation, development and production of biological medicines based on the human microbiome to treat intestinal alterations. Currently, its product MBK-01 has just finished a Phase III clinical trial for *C. difficile* infection. MBK01 is dispensed in capsules made of lyophilised faecal microbiota from previously certified faecal donors. The use of capsules has demonstrated to be an effective (> 80%) and non-invasive alternative with few side effects [37,38].

Although FMT procedures have been successfully conducted with samples from a single donor, there is a growing preference for mixing several donors to increase the microbial diversity of the samples. To date, it is still unknown if using multiple donors could be more effective than the use of just one for each patient, and this is certainly a key point to be addressed by the manufacturers.

In the present study, 16S rDNA massive sequencing was performed on consecutive faecal samples from certified donors at Mikrobiomik Company to evaluate whether knowledge of microbiota composition is necessary to improve donor selection criteria. Donor selection based on a healthy microbiota has always been the main goal of FMT; however, it has been limited by the inability to define what constitutes a healthy microbiota. Nevertheless, it is generally accepted that a stable and diverse microbiota, characterised by a high number of different species (alpha diversity) and richness in the abundance of taxa (beta diversity), is correlated with a healthy gut state.

The results obtained in this study showed that most of the faecal donors had similar alpha diversity values measured by the Chao 1 and Shannon indices. However, donor 23 had a low Shannon index, suggesting that the microbiota of donor 23 may lack some of the variety found in the other samples. In contrast, donors 17 and 19 had a richer and more diverse gut microbiota than the others. Furthermore, the comparation of microbiota diversity across donors (beta diversity) using the Bray–Curtis index indicated that each donor has a distinct microbial ecosystem but with some similarities, except donors 23 and 29 whose microbial composition differed from the other donors. Additionally, some variability was observed among the stool samples collected from the same donor, indicating fluctuations in their microbiota composition over time. These fluctuations can be consequences of different factors, such as diet, environment and overall health [39]. Similar results had already been documented in the studies of the Human Microbiome Project, where inter-individual variation in the gut microbiome proved to be specific and functionally relevant [40].

In our study, we examined the taxonomy of the gut microbiota of the donors using 16S rDNA sequencing, paying special attention to the phyla and genera composition. We found that the phylum *Bacillota* (64.5%) was the most abundant in all donors, followed by *Bacteroidota* (16.7%), *Actinomycota* (6.7%), *Verrucomycota* (6.7%) and *Pseudomonadota* (2.9%), with some inter-donor and intra-donor variations. Other bacterial phyla were also detected in the donor samples with lower abundances. While it is widely recognised that the composition of gut microbiota can vary significantly across different individuals, there are certain characteristics that are typically associated with a balanced gut microbiota. These include the prominent representation of members of the phyla *Bacillota* and *Bacteoroidota* and a lower abundance of members of the phyla *Pseudomonota*, *Actynomycota* and *Verrumicota* [41,42]. We also observed a notable variability in the phyla distribution among samples from the same donor taken at different time points.

At the genus level, the main bacterial genus in all donors were *Agathobacter* (13.35%), *Eubacterium* (11%) and *Bacteroides* (6.71%), except in donor number 13 where *Prevotella* was the dominant genus. Despite the variability in the genus density between donors and within the same donor, it was possible to define a core microbiome consisting of *Agathobacter, Eubacterium, Bacteroides, Clostridia UCG-014* and *Akkermansia*. The predominance of the genus *Agathobacter* in healthy adults has been previously documented. In 2021, a large-scale study analysed the gut microbiota composition using metagenomic analysis and found that the most predominant species in adult samples was *Agathobacter rectalis*, traditionally known as *Eubacterioum rectale* [43].

The concept of a core microbiota is based on the prevalence of certain bacterial genus in donor samples. Studies from the Human Microbiome Project had analysed numerous samples from healthy individuals and found variations in microbial composition, but metabolic pathways remained conserved [40,44]. For this reason, it would be more insightful to focus on the core metabolome rather than the core microbiota, understanding this as the common metabolic functions shared by all individuals. The genus *Agathobacter*, specifically the species *A. rectalis*, along with some species from the genus *Eubacterium* constitute an important and abundant group of *Firmicutes* related to the production of butyrate, a short-chain fatty acid with positive effects on the microbiota and the host [45]. *Bacteroides* spp. (*Bacteroidetes* phylum) is one of the most prominent genera in the microbiota and plays a key role in human gut health. This genus is crucial for the digestion of complex polysaccharides and helps regulate the immune response by producing two short-chain fatty acids, propionate and acetate, which have anti-inflammatory effects [46]. Additionally, *Akkermansia* spp. (*Verrumicota* phylum), particularly *Akkermansia muciniphila*, has been extensively studied in recent years as a promising probiotic due to its benefits in metabolic and intestinal diseases, although its precise function in gut health remains somewhat ambiguous. *A. muciniphila* adheres to the intestinal mucosa and feeds on mucus, which helps maintain the integrity of the gut barrier [47]. In contrast to these genera that constitute the core microbiome, there is limited information available on the genus *Clostridia UCG-014.* In conclusion, the core microbiome, which includes the genus *Agathobacter, Eubacterium, Bacteroides, Clostridia* UCG-014 and *Akkermansia*, likely reflects a microbial community that contributes to intestinal health in our 10 donors, with uncertainty as to whether these results can be extrapolated to other donors.

To complete this investigation, we represented the genera composition of the ten certified donors in a heatmap to illustrate and compare the characteristic microbiota patterns of each donor. These results further emphasise that each donor has a unique gut composition with intra-individual variations. Notably, only donors 23 and 29 exhibited compositional patterns that remained more stable over time.

According to the characteristics of a healthy microbiota mentioned above (stability and diversity), only donor 29 has a compatible microbiota profile. In contrast, the other donors exhibited significant fluctuations in their microbiota composition, highlighting the dynamic nature of the gut microbiota. The concept that a healthy microbiome can be defined by specific populations of certain bacteria is too limited [48]. It is essential to include metabolomic studies to determine whether these differences in microbial composition are translated into variations in ecosystem functionality.

## 5. Conclusions

The results obtained in the present work provide a descriptive analysis of the composition and stability of 10 microbiota-certified donors. While most of them exhibited similar microbial diversity, one individual showed diversity and richness that could be associated with a healthier microbiota, although the criteria for normality have not yet been defined. We know that one limitation of this study is that we lack individual effectiveness data for each certified donor, as not all the samples that we studied herein were administered to rCDI patients. However, as mentioned above, the role of a “super-donor” has not been described in *C. difficile* infection patients undergoing this treatment. Our conclusion is that the extensive knowledge of the microbiota composition is not essential for stool donor certification, since, like all healthy individuals, each one has its own pattern, without interfering with the result of the FMT. Since different microbiota patterns can have the same metabolic functionality, studying molecules such as short-chain fatty acids could be useful in the future; however, this aspect has yet to be explored in healthy donors. Currently, the inability to incorporate microbial and metabolic analyses of donors still limits their certification to lifestyle factors and the absence of known pathogens and diseases.

## Figures and Tables

**Figure 1 diagnostics-14-02635-f001:**
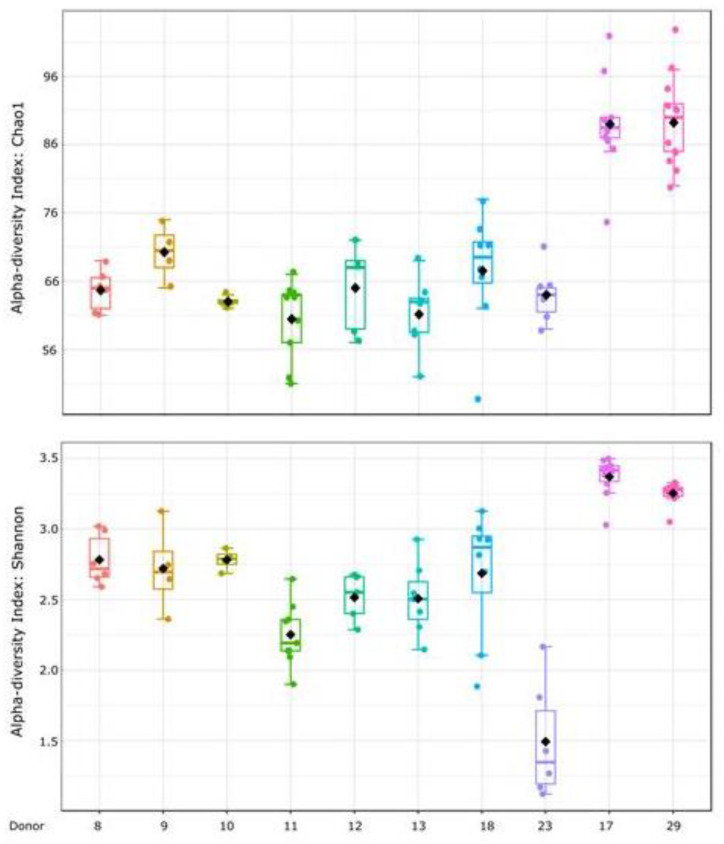
Alpha diversity values for each donor’s samples regarding the number and abundance of amplicon sequence variants (ASVs). (**A**) Chao1 index for richness in genera species (Kruskal–Wallis statistic 50.999, *p*-value < 0.001) and (**B**) Shannon index for abundance distribution of the genera (Kruskal–Wallis statistic 58.989, *p*-value < 0.001).

**Figure 2 diagnostics-14-02635-f002:**
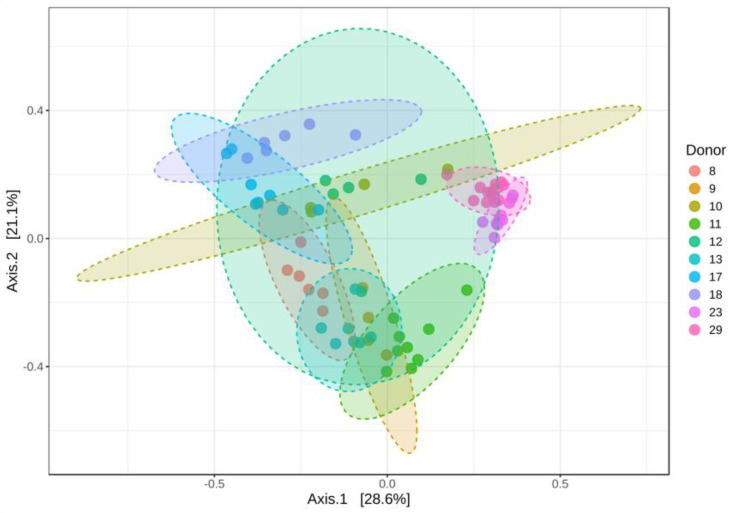
Beta diversity values for each donor and sample. Principal coordinate analyses (PCoA) of the Bray–Curtis comparison (F-value: 22.98; R-squared: 0.77; *p*-value > 0.001). The ellipse size represents microbial composition variations among samples from each faecal donor.

**Figure 3 diagnostics-14-02635-f003:**
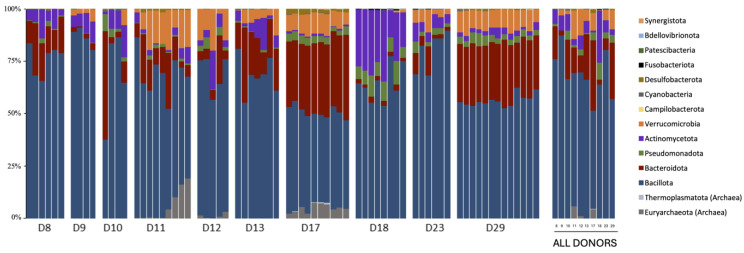
Phyla distribution in the different samples of each donor. The most abundant phyla are Bacillota and Bacteroidota. Median values for each donor were obtained from their samples and are represented as individual values.

**Figure 4 diagnostics-14-02635-f004:**
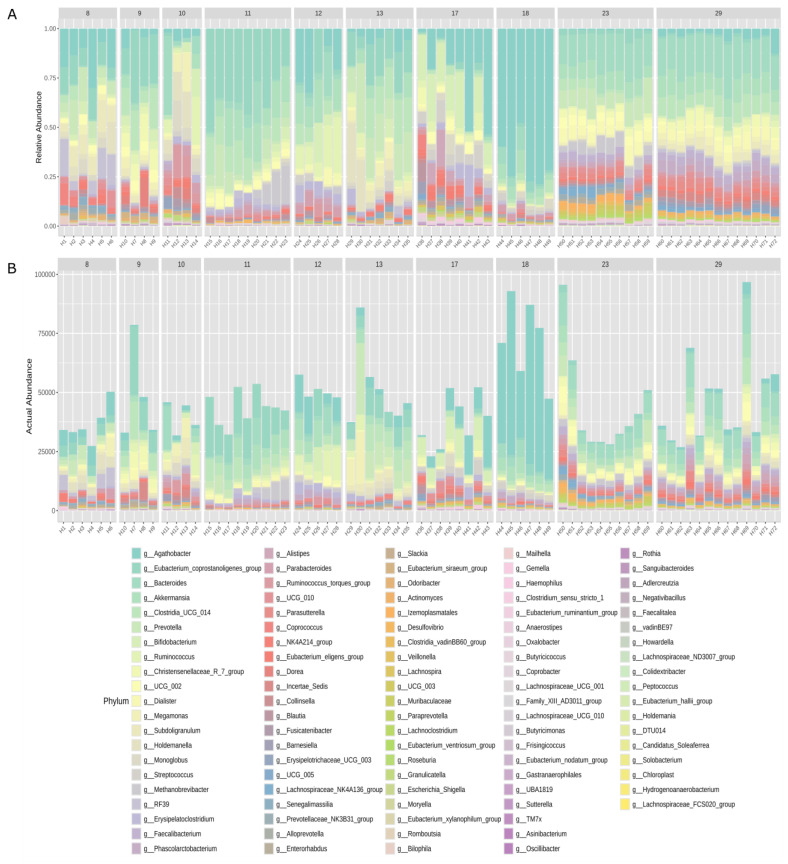
Microbial community structure in certified faecal donors. Relative (**A**) and total abundance (**B**) of the genera identified by each donor and sample are shown.

**Figure 5 diagnostics-14-02635-f005:**
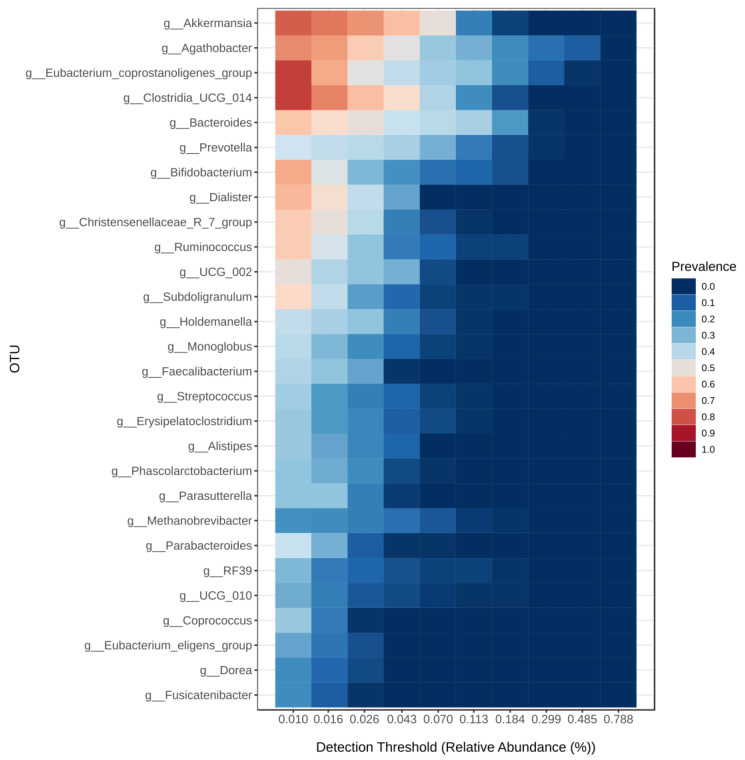
Microbiota core of certified faecal donors defined by the relative abundance (%) of the main genera (*X*-axis) and their prevalence, represented by a colour scale (red: high prevalence, light blue: medium or low prevalence, dark blue: very low prevalence).

**Figure 6 diagnostics-14-02635-f006:**
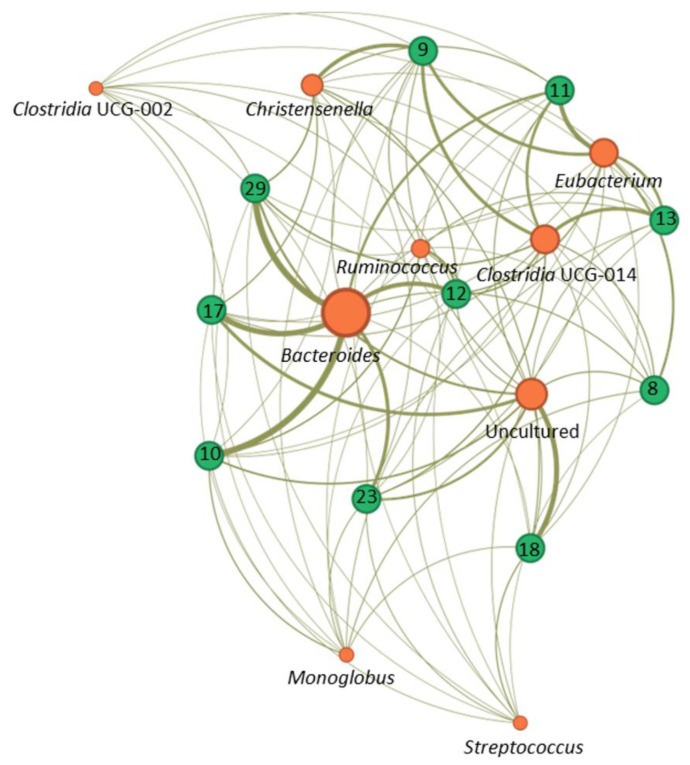
Clustering of faecal microbiota composition in certified faecal donors. Each column represents a sample from the 10 donors evaluated, and each row represents a bacterial genus identified in the microbiota by 16S rDNA sequencing. Hierarchical clustering was applied using a Euclidean distance matrix and the Ward linkage method. The colour scale of the heatmap indicates the relative abundance of a specific bacterial genus corresponding to a sample. Donors are represented in green, and the numbers corresponded to the number connections between genera.

**Table 1 diagnostics-14-02635-t001:** Main characteristics of the faecal donors.

Donor Number	Age	Gender	Number of Samples	Body Mass Index
D08 ^1^	30	Male	6	27.76
D09	31	Female	4	20.31
D10	23	Male	4	22.16
D11	23	Male	9	22.52
D12	43	Female	5	19.72
D13	31	Male	7	24.03
D17	31	Male	10	21.53
D18	24	Male	8	18.69
D23	29	Female	6	19.50
D29	18	Male	13	20.40

^1^ D: donor.

## Data Availability

Sequences are deposited under the number PRJNA1117606.

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
