# Peer review of "Should the Faecal Microbiota Composition Be Determined to Certify a Faecal Donor?"

_diagnostics, 2024, doi:10.3390/diagnostics14232635_

Round 1
Reviewer 1 Report
Comments and Suggestions for Authors
Comments:
Title: Is it essential to know the composition of the faecal microbiota to certify a faecal donor?"
Title should be change, it could be more clearly defined.
Abstract Section:
1) Background/Objectives: FMT could benefit from a brief explanation of why understanding microbiota composition is important. Adding context about the implications of donor selection on patient outcomes could strengthen the rationale.
2) Methods: The authors mention "massive sequencing of 16S rDNA" but do not specify the sequencing technology or platform used in this study. Clarifying this detail would be beneficial for readers to fully understand the methodology.
3-1) All keywords should be checked according to MeSH.
3-2) I think the authors should consider adding terms that reflect the specific findings or methodologies discussed, such as '16S rDNA sequencing' or 'microbial diversity metrics.'
Introduction Section:
The introduction is too long and quite dense, covering many aspects of FMT. It may benefit from a clearer focus on the main objectives of the study. Consider streamlining the content to emphasize the most relevant points that directly lead to the research question.
Additionally, while the use of 16S rDNA sequencing is mentioned, there is not enough detail on how this methodology addresses the identified gaps in donor certification. A brief explanation of why this method is particularly suitable for the study would be beneficial. The introduction touches on the limitations of donor validation but the authors could elaborate on why current practices are insufficient. Discussing specific challenges in current methodologies could provide a clearer picture of the problem being addressed.
Materials and Methods Section:
1) There is no explanation provided regarding which methods were considered for data analysis, significance levels and the reasons for choosing those methods.
2) Ethical considerations are not found in the manuscript. Please check and confirm it.
3) The authors mention that selective amplification of the V3 and V4 regions was performed; however, they do not provide information on the primers used, PCR conditions, or any controls included in the amplification process. This information is essential for assessing the reliability of the results.
Comments on the Quality of English LanguageThe English could be improved to more clearly express the research.
Author Response
Comment 1: Title: Is it essential to know the composition of the faecal microbiota to certify a faecal donor?" Title should be change, it could be more clearly defined.
Response 1: We appreciate the reviewer's comment, we hope that the new modification of the title is appropriate.
Comment 2: 1) Background/Objectives: FMT could benefit from a brief explanation of why understanding microbiota composition is important. Adding context about the implications of donor selection on patient outcomes could strengthen the rationale.
Response 2: We agree with the observation, and a new phrase has been introduced in the new version.
Coment 3: Methods: The authors mention "massive sequencing of 16S rDNA" but do not specify the sequencing technology or platform used in this study. Clarifying this detail would be beneficial for readers to fully understand the methodology.
Answer 3: It was included
Comment 4: All keywords should be checked according to MeSH.
Answer 4: It has been corrected.
Coment 5: I think the authors should consider adding terms that reflect the specific findings or methodologies discussed, such as '16S rDNA sequencing' or 'microbial diversity metrics.'
Answer 5: It has been corrected
Comments 6: The introduction is too long and quite dense, covering many aspects of FMT. It may benefit from a clearer focus on the main objectives of the study. Consider streamlining the content to emphasize the most relevant points that directly lead to the research question.
Answer 6: We agree with the author, but we also consider that all the points in the introduction are necessary in order to focus the subject of the paper and to enable the reader who is not an expert on microbiota to understand.
Comment 7: Additionally, while the use of 16S rDNA sequencing is mentioned, there is not enough detail on how this methodology addresses the identified gaps in donor certification. A brief explanation of why this method is particularly suitable for the study would be beneficial. The introduction touches on the limitations of donor validation but the authors could elaborate on why current practices are insufficient. Discussing specific challenges in current methodologies could provide a clearer picture of the problem being addressed.
Answer 7: A comment has been introduced in the new version as suggested by the reviewer.
Comment 8: 1) There is no explanation provided regarding which methods were considered for data analysis, significance levels and the reasons for choosing those methods.
Results 8: As described in the M&M section, we have used the default parameters of each platform of analysis, as they are commonly used in our system.
Comment 9: 2) Ethical considerations are not found in the manuscript. Please check and confirm it.
Answer 9: We apologize for the mistake, the ethical references has been included in teh new version.
Comment 10: 3) The authors mention that selective amplification of the V3 and V4 regions was performed; however, they do not provide information on the primers used, PCR conditions, or any controls included in the amplification process. This information is essential for assessing the reliability of the results.
Answer 10: We apologize for the missing data, that has been included in the new version.
Reviewer 2 Report
Comments and Suggestions for Authors
The writing of the names of the phyla should be homogeneous and should be in italics.
There should also be some discussion of the bioethical implications of faecal matter transfer.
Work such as that presented by the authors opens a door not only to treat patients with C.difficile problems, but also alternatives to reduce antimicrobial resistance and to control diseases such as Chron's disease.
The authors state that there is still a long way to go before fecal material transplantation can be seen as a treatment in some countries, as lifestyle, age and gender are very diverse in the world.
Author Response
Comment 1: The writing of the names of the phyla should be homogeneous and should be in italics.
Response 1: The phyla names have been revised and corrected. Please note that in the figures, we cannot modify, what is shown is the original result from the web pages.
Comment 2: There should also be some discussion of the bioethical implications of faecal matter transfer.
Response 2: We understand the reviewer's concern on the bioethical aspect, but in this case we are dealing with a drug with clear therapeutic indications, so the only bioethical discussion would be how to match donors with recipients, but for the moment this has not been developed. We ourselves tried to do a donor/recipient selection test and it was impossible (see DOI: 10.3389/fimmu.2021.683387). In case the reviewer refers to another issue, please let us know and we will be pleased to discuss it in the new version of the manuscript.
Comment 3: Work such as that presented by the authors opens a door not only to treat patients with C. difficile problems, but also alternatives to reduce antimicrobial resistance and to control diseases such as Chron's disease
Response 3: We agree with the referee, however, the only clear indication at present is recurrence of C. dificille. We will of course continue research to elucidate the application of this technique in other pathologies.
Comment 4: The authors state that there is still a long way to go before fecal material transplantation can be seen as a treatment in some countries, as lifestyle, age and gender are very diverse in the world.
Response 4: Thanks for the comment
Round 2
Reviewer 1 Report
Comments and Suggestions for Authors
Accept
Comments on the Quality of English LanguageIt is OK